

# The distribution and conservation of areas with microendemic species in a biodiversity hotspot: a multi-taxa approach

Helder F. P. de Araujo[1], Célia C. C. Machado[2] and José Maria Cardoso da Silva[3]

[1] Department of Biosciences, Federal University of Paraíba, Areias, Paraíba, Brazil
[2] Center of Applied Biological and Social Sciences, State University of Paraíba, João Pessoa, Paraíba, Brazil
[3] Department of Geography and Sustainable Development, University of Miami, Coral Gables, Florida, United States of America

## ABSTRACT

**Background:** Microendemic species are species with very small geographic distributions (ranges). Their presence delimitates areas with microendemic species (AMs), denoting a spatial unit comprising at least one population of at least one microendemic species. AMs are assumed to be distributed distinctively and associated with specific ecological, historical, and anthropogenic attributes. However, the level of influence of these factors remains unclear. Thus, we studied the distribution patterns of microendemic species within the Brazilian Atlantic Forest to (a) identify the region's AMs; (b) evaluate whether ecological (latitude, altitude, distance from the coastline), historical (climate stability), and anthropogenic (ecological integrity) attributes distinguish AMs from non-AMs; and (c) assess the conservation status of the Atlantic Forest's AMs.

**Methods:** We mapped the ranges of 1,362 microendemic species of angiosperms, freshwater fishes, and terrestrial vertebrates (snakes, passerine birds, and small mammals) to identify the region's AMs. Further, spatial autoregressive logit regression models were used to evaluate whether latitude, altitude, distance from the coastline, Climate Stability Index, and ecological integrity can be used to discern AMs from non-AMs. Moreover, the AMs' conservation status was assessed by evaluating the region's ecological integrity and conservation efforts (measured as the proportion of AMs in protected areas).

**Results:** We identified 261 AMs for angiosperm, 205 AMs for freshwater fishes, and 102 AMs for terrestrial vertebrates in the Brazilian Atlantic Forest, totaling 474 AMs covering 23.8% of the region. The Brazilian Atlantic Forest is a large and complex biogeographic mosaic where AMs represent islands or archipelagoes surrounded by transition areas with no microendemic species. All local attributes help to distinguish AMs from non-AMs, but their impacts vary across taxonomic groups. Around 69% of AMs have low ecological integrity and poor conservation efforts, indicating that most microendemic species are under threat. This study provides insights into the biogeography of one of the most important global biodiversity hotspots, creating a foundation for comparative studies using other tropical forest regions.

Corresponding author
José Maria Cardoso da Silva,
jcsilva@miami.edu

# INTRODUCTION

Microendemic species occupy very small geographic distributions or ranges (*Nogueira et al., 2010*; *Hobohm, 2013*; *Silva et al., 2019*). They belong to various organism groups, including angiosperms, insects, and vertebrates (*Kruckeberg & Rabinowitz, 1985*; *Wamelink, Goedhart & Frissel, 2014*). If ranges are geographic expressions of a species' ecological niche (*Peterson et al., 2011*), then microendemic species occupy the narrowest of ecological niches (*Wamelink, Goedhart & Frissel, 2014*), rendering them more sensitive to disturbances (*Lozada et al., 2008*) and highly dependent on habitat integrity for survival (*Wulff et al., 2013*; *Caesar, Grandcolas & Pellens, 2017*).

Microendemic species are not distributed randomly; rather, their ranges comprise unique locations (*Kruckeberg & Rabinowitz, 1985*; *Wulff et al., 2013*; *Pimm et al., 2014*; *Caesar, Grandcolas & Pellens, 2017*; *Silva et al., 2019*), termed areas with microendemic species (AMs), that is, spatial units containing at least one population of at least one microendemic species. Identifying AMs is important because they are considered unique from a biodiversity perspective and a top priority for conservation efforts (*Kruckeberg & Rabinowitz, 1985*; *Pressey, Johnson & Wilson, 1994*; *Silva et al., 2019*).

AMs can be part of a "cradle," where young species have evolved, or a part of a "museum," where old species have survived long after disappearing from other parts of their ranges, whether due to natural or anthropogenic environmental changes (*Kier et al., 2009*; *Albert, Petry & Reis, 2011*; *Rahbek et al., 2019*). Because of these roles, AMs are assumed to be characterized by unique attributes compared to non-AMs. However, the relative contributions of these factors in explaining current AM distribution patterns still need clarification (*Hobohm, 2013*).

Studies aiming to distinguish places with endemic species from those without have focused on two particular sets of attributes: ecological and historical (*Fjeldså, Lambin & Mertens, 1999*; *Harrison & Noss, 2017*), where the former represent contemporary environmental conditions and the latter the signature of past ecological conditions. More recently, biogeographers have also considered the impact of human activities (or anthropogenic attributes) on places and their biotas, as most terrestrial and marine ecosystems face at least some degree of human disturbance (*Halpern et al., 2012*; *Williams et al., 2020*). It is possible that ecological, historical, and anthropogenic attributes synergistically influence the probability of a place maintaining viable populations of one or more microendemic species (hereafter AM probability); thus, they should be evaluated simultaneously rather than individually. Furthermore, assessing how these attributes influence AM probability requires a multi-taxa approach, as different taxonomic groups are expected to respond diversely to these attributes due to their distinct traits and habitat requirements (*Pacifici et al., 2017*; *Beissinger & Riddell, 2021*; *Green et al., 2022*).

Three ecological (latitude, distance from the coastline, and altitude), one historical (long-term climatic stability) and one anthropogenic attribute (ecological integrity,

measured by the percentage of a place covered by natural vegetation) are thought to be leading candidates in distinguishing AMs from non-AMs. First, latitude is expected to have a negative association with AM probability because low-latitude locations house more species having small ranges than high-latitude places, primarily due to a combination of narrow species' ecological tolerance, high speciation rates, and long-term spatiotemporal variation in precipitation regimes (*Saupe et al., 2019*). Similarly, AM probability is predicted to decrease with the distance from the coastline because coastal places tend to be more heterogeneous and productive, as well as less seasonal than inland places (*Jenkins, Pimm & Joppa, 2013*; *Hobohm, 2013*; *Pimm et al., 2014*; *Acevedo & Sandel, 2021*). Conversely, AM probability is predicted to increase with altitude because high-altitude places are typically more isolated, smaller and comprised of more complex topographies than low-altitude places, leading to sharp habitat changes in a relatively small area and enabling the origin and survival of microendemic species (*Steinbauer et al., 2016*; *Rahbek et al., 2019*). Long-term climatic stability is predicted to have a positive association with AM probability because places that remained stable during the recent recurrent large-scale climatic changes are more likely to have served as "ecological refugia" for narrow niche species, including microendemic species (*Haffer, 1985*; *Fjeldså, Lambin & Mertens, 1999*; *Ravelo et al., 2004*; *Kier et al., 2009*; *Harrison & Noss, 2017*; *Rahbek et al., 2019*). Finally, AM probability is predicted to increase along with a location's ecological integrity because places dominated by native vegetation are more likely to maintain healthy microendemic species populations than those with low ecological integrity (*Pimm et al., 2014*).

For seven reasons, the Atlantic Forest, one of the largest South American biodiversity hotspots (*Mittermeier et al., 2005*), is a natural laboratory for studying AMs and the attributes setting them apart from non-AMs. First, the region covers around 1.4 million km$^2$ and harbors an unparalleled density of local to regional endemic species crowded into a large and environmentally heterogeneous region (*Tabarelli et al., 2005*). Second, the Atlantic Forest occupies a large latitudinal extent (ca. 25 degrees) along the South American Atlantic coastline from the Rio Grande do Norte in Northeastern Brazil to Southern Brazil (*Galindo-Leal & Camara, 2003*; *Instituto Brasileiro. de Geografia e Estatística, 2019*). Third, the Atlantic Forest has a large longitudinal extent, occupying vast areas from the coast to Central Brazil, Northeastern Argentina (Misiones), and Southwestern Paraguay (*Galindo-Leal & Camara, 2003*). Fourth, the Atlantic Forest has a complex topography, with altitudes ranging from 0 to 2,892 m above sea level, forming ecological gradients that influence species distribution (*Goerck, 1999*; *Silva, Sousa & Castelletti, 2004*). Fifth, the Atlantic Forest has experienced large-scale palaeoecological changes in the last 3–4 million years, with some places being more climatically stable than others (*Carnaval & Moritz, 2008*; *Peres et al., 2020*). Sixth, the Atlantic Forest has lost more than 80% of its original native vegetation, and what is left is unevenly distributed across its sub-regions (*Ribeiro et al., 2009*; *Silva et al., 2016*).

This article has three primary goals. The first is to map the ranges of the microendemic species of three taxonomic groups (angiosperms, freshwater fishes, and terrestrial

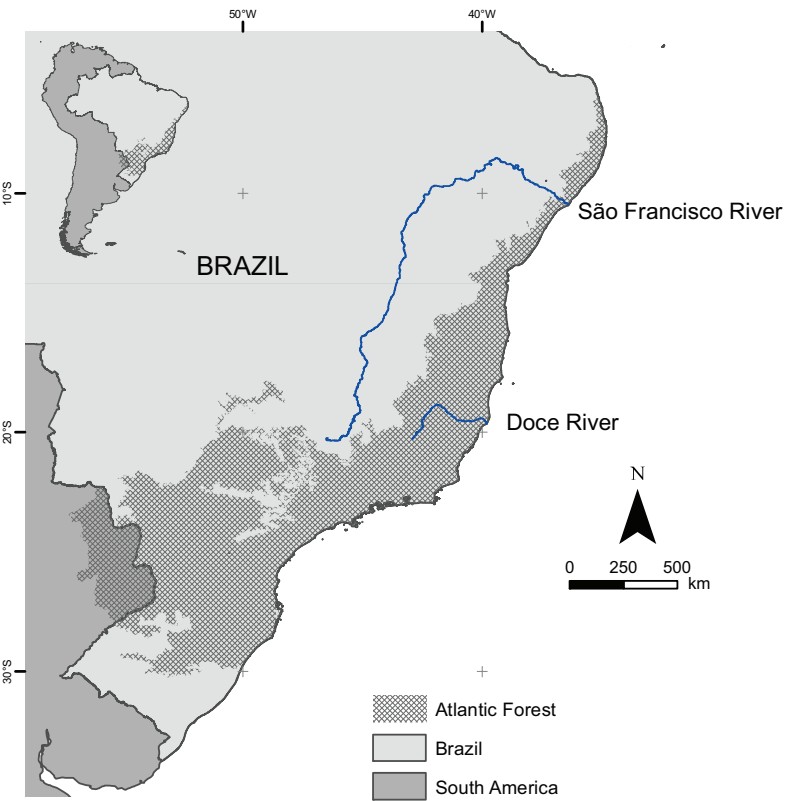

**Figure 1 Boundaries of the Atlantic Forest in Brazil and South America.**

vertebrates) to identify the Atlantic Forest's AMs. The second goal is to apply spatial autoregressive logit regression models to assess the influence of latitude, altitude, distance from the coastline, climate stability, and ecological integrity on AM probability. Finally, the third is to evaluate the conservation status of the Atlantic Forest's AMs and propose general recommendations to safeguard the region's microendemic species. This study is on the interface between biogeography and conservation science, as it not only generates new knowledge on the biogeography of a major global biodiversity hotspot, but it also pinpoints important places where conservation actions should be directed to avoid the extinction of microendemic species, laying the groundwork for future comparative research on other tropical forest hotspots.

## MATERIALS AND METHODS

### Study area

Our analysis is restricted to the Brazilian Atlantic Forest (Fig. 1), as defined by the Brazilian Institute of Geography and Statistics (IBGE, from its Brazilian name: Instituto Brasileiro de Geografia e Estatística) (*Instituto Brasileiro. de Geografia e Estatística, 2019*). The IBGE's limits differ from those of other studies (*Silva & Casteleti, 2003*; *Silva, Sousa & Castelletti, 2004*; *Peres et al., 2020*) because they exclude humid forests and tropical dry forests found in the Caatinga and Cerrado regions (*de Araujo et al., 2022*). Atlantic Forests areas outside

Brazil (*e.g.*, Argentina and Paraguay) were not included in this analysis because comparative and reliable taxonomic and biogeographical data were unavailable.

## Species datasets

We mapped the ranges of microendemic species (*i.e.*, species with ranges smaller than 10,000 km$^2$) of angiosperms, freshwater fishes, and terrestrial vertebrates (snakes, small mammals, and passerine birds) because comparable taxonomic and biogeographical reliable data were available for them. For angiosperms and freshwater fishes, we used the databases gathered by *Silva et al. (2019)* and *Nogueira et al. (2010)*, respectively, whereas for snakes and mammals, we selected microendemic species from the datasets generated by *Barbo, Nogueira & Sawaya (2021)* and *Dalapicolla et al. (2021)*, respectively. For passerine birds, we first selected microendemic species from the list of Brazilian Atlantic Forest endemic species produced by *Silva, Sousa & Castelletti (2004)*, but we updated the species ranges using recent publications (*Bello et al., 2017*; *Rodrigues et al., 2019*), as well as new and validated (*i.e.*, represented by specimens, voices, or pictures) records from Wikiaves (www.wikiaves.com.br) and Global Biodiversity Information Facility (www.gbif.org).

## Areas of microendemism

We used QGIS to create detailed maps of all microendemic species within each taxonomic group in the Brazilian Atlantic Forest. Then, we superimposed these maps with a map dividing the Atlantic Forest into 2,243 equal-sized hexagons (ca. 630 km$^2$) and counted the microendemic species recorded within them. Any hexagon containing at least one record of a microendemic species was designated an AM.

## Explanatory variables

We have gathered data on five local attributes for all 2,243 hexagons: latitude, altitude, distance from the coastline, Climate Stability Index (CSI), and ecological integrity, all of which were collected from publicly available digital databases. Latitude is represented by the absolute latitude value (in decimal degrees) of the hexagon's centroid. Elevation is the average elevation (in m) within the hexagon. This information came from the WorldClim 2.1 platform (*Fick & Hijmans, 2017*), with a resolution of 2.5 arc minutes or ca. 5 km$^2$ (https://www.worldclim.org/). The distance from the hexagon to the nearest coast was calculated using a global database provided by NASA's Ocean Biology Processing Group (https://oceancolor.gsfc.nasa.gov/docs/distfromcoast/). The CSI represents a hexagon's climate stability index average since the Pliocene (*Herrando-Moraira et al., 2022*). The CSI measures the variability of 14 bioclimatic variables using nine general circulation climate change models over four periods, available from WorldClim at a resolution of ca. 5 km. The final CSI maps were obtained by summing the standard deviations of the variables selected and the normalized subsequent outputs (*Herrando-Moraira et al., 2022*), and it ranged from 0 (low climatic fluctuations) to 1 (high climatic fluctuations). Finally, we measured each hexagon's ecological integrity by calculating the percentage of its area

covered with natural vegetation using the 2021 Annual Land Use and Land Cover map for Brazil produced by Mapbiomas (https://brasil.mapbiomas.org).

## Statistical analyses

Before any analysis, we evaluated whether the explanatory variables present multicollinearity by estimating their variance inflation indices (VIFs) using the command estat vif in Stata (*StataCorp, 2017*). All explanatory variables had VIFs below five, indicating no multicollinearity, so they were added to all models.

We used spatial autoregressive logit regression models to test the hypotheses of associations between the five local attributes simultaneously and the AM probability of each taxonomic group. Models were built using the non-linear two-stage least squares (N2SLS) estimator in the spatbinary command in Stata (*Spinelli, 2022*). Spatial models differ from aspatial models because they consider and model the spatial relationships and dependencies among data points by considering their geographic distance from each other. An inverse geographic distance matrix generated using the spmatrix command in Stata (*StataCorp, 2017*) was used in the model, and the models' rho coefficients were used to assess spatial autocorrelation in the dataset. Further, Hansen's test for overidentification was used to evaluate whether the number of explanatory variables was greater than the number of parameters to be estimated (*Spinelli, 2022*). The coefficients of a spatial logit regression show the direction (positive or negative) of the relationship between each attribute and AM but not the attribute's impacts on AM probability. To assess such impacts, we used the command spatbinary_impact in Stata (*Spinelli, 2022*) to generate the elasticities (*i.e.*, the percent variation in the response variable in relation to the 1% variation in an explanatory variable if the rate remained constant) of each explanatory variable of each model. Impacts can be direct, indirect, and total, where the former measures the hexagon's predicted contributions to its probability of a positive outcome (*i.e.*, being an AM), whereas the indirect impact measures the predicted impact of the other hexagons' contributions to a hexagon's probability of being an AM and the total impact is the sum of direct and indirect impacts.

## Conservation status of areas of microendemic species

To assess the conservation status of each AM, we combined the indicator of ecological integrity with an indicator of conservation effort. To measure conservation efforts, we overlaid the 2021 map of all protected areas in Brazil's Protected Area National Register (https://cnuc.mma.gov.br/) atop the hexagons using QGIS and estimated the percentage within protected areas. We used 50% as the minimum ecological integrity and conservation effort required to maintain healthy populations of microendemic species within an AM (*Wilson, 2017*). By using this threshold, we classified the Atlantic Forest's AMs into four conservation statuses: (a) AMs with high ecological integrity and high conservation effort; (b) AMs with high ecological integrity and low conservation effort; (c) AMs with low ecological integrity and high conservation effort; and (d) AMs with low ecological integrity and low conservation effort.

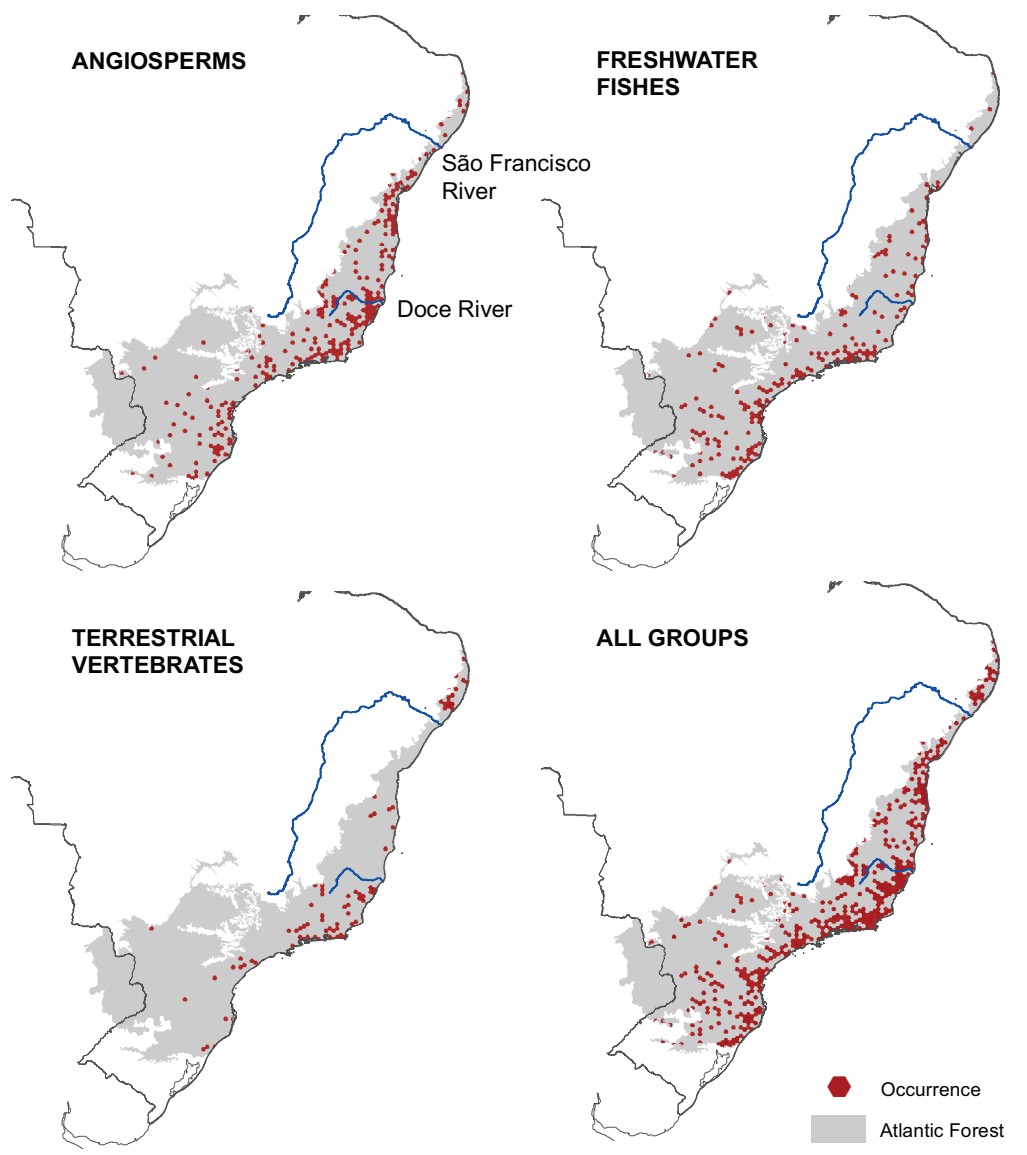

**Figure 2 Geographic distribution of areas with microendemism for angiosperms, freshwater fishes, terrestrial vertebrates and all three groups combined in the Brazilian Atlantic Forest.**

## RESULTS

### Areas of microendemism

We mapped the ranges of 1,362 microendemic species, of which 994 were angiosperms, 321 were freshwater fishes, and 47 were terrestrial vertebrates. Among the latter, 11 were snakes, 28 were small mammals, and eight were passerine birds. By analyzing these ranges, we identified 261 AMs for angiosperm, 205 AMs for freshwater fishes, and 102 AMs for terrestrial vertebrates (Fig. 2), representing 13.6%, 10.3%, and 4.8% of the Atlantic Forest's total area, respectively. Further, when combining all taxonomic groups, the number of AMs is 474, corresponding to 23.8% of the Atlantic Forest's total area (Fig. 2). AMs are
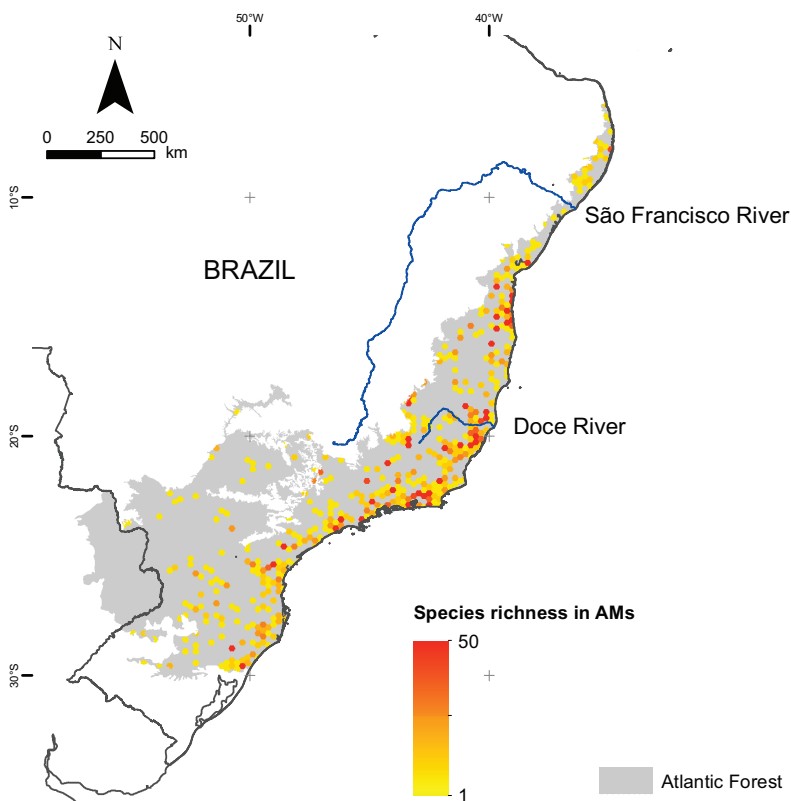

**Figure 3 Species richness of microendemic species in the Brazilian Atlantic Forest.**

found everywhere across the region (Fig. 2), with the total number of microendemic species in each AM varying from 1 to 50 (Fig. 3), but 50.8% of AMs harbor only one species. Regardless, AMs with high species richness are found throughout the region.

## Association between areas of microendemism and local attributes

The association between AM probability and local attributes differed across taxonomic groups (Table 1). For angiosperms, AM probability increased with climate stability and ecological integrity but decreased with the distance from the coastline. Further, the AM probability for freshwater fishes increased with latitude and ecological integrity but decreased with the distance from the coastline. Finally, for terrestrial vertebrates, AM probability increased with altitude and ecological integrity but decreased with latitude, distance from the coastline, and climate stability. In addition, Hansen's tests for all three spatial regression models (Table 1) were not significant (seed plants: $\chi^2 = 6.82$, df = 9, $p = 0.65$; freshwater fishes: $\chi^2 = 1.64$, df = 9, $p = 0.99$; terrestrial vertebrates: $\chi^2 = 2.63$, df = 9, $p = 0.97$); thus, all explanatory variables used in the models are valid (Table 1).

The rho values show that AMs do not show spatial autocorrelation (Table 1), and consistent with this result, no measure of indirect impact was significant across the three regression models (Tables 2–4). For angiosperms, a 1% increase in distance from the coastline and ecological integrity resulted in a 1.7% reduction and a 0.7% increase in AM

**Table 1 Relationship between AM probability in the Brazilian Atlantic Forest and five local attributes (latitude, altitude, distance from the coastline, climate stability, and ecological integrity) across different taxonomic groups.**

| Group/Attributes | Coefficient | Robust S.E. | z | p |
|---|---|---|---|---|
| **Angiosperms** | | | | |
| Latitude | 0.01 | 0.03 | 0.48 | 0.63 |
| Altitude | 0.00 | 0.00 | 1.77 | 0.08 |
| Distance from the coastline | −0.01 | 0.00 | −5.05 | 0.00 |
| Climate stability index | 4.50 | 2.06 | 2.19 | 0.03 |
| Ecological integrity | 0.02 | 0.00 | 6.96 | 0.00 |
| Constant | −2.44 | 0.66 | −3.70 | 0.00 |
| Rho | 0.32 | 0.19 | 1.65 | 0.10 |
| **Freshwater fishes** | | | | |
| Latitude | 0.09 | 0.03 | 2.79 | 0.01 |
| Altitude | 0.00 | 0.00 | −1.04 | 0.30 |
| Distance from the coastline | −0.01 | 0.00 | −4.14 | 0.00 |
| Climate stability index | 3.46 | 2.81 | 1.23 | 0.22 |
| Ecological integrity | 0.02 | 0.00 | 3.95 | 0.00 |
| Constant | −4.49 | 0.98 | −4.60 | 0.00 |
| Rho | −0.11 | 0.25 | −0.44 | 0.66 |
| **Terrestrial vertebrates** | | | | |
| Latitude | −0.093 | 0.04 | −2.18 | 0.03 |
| Altitude | 0.002 | 0.00 | 4.96 | 0.00 |
| Distance from the coastline | −0.020 | 0.00 | −4.30 | 0.00 |
| Climate stability index | −8.141 | 2.73 | −2.99 | 0.00 |
| Ecological integrity | 0.023 | 0.01 | 3.83 | 0.00 |
| Constant | 0.493 | 0.81 | 0.61 | 0.54 |
| Rho | 0.06 | 0.16 | 0.35 | 0.73 |

probability, respectively (Fig. 2). Although the total impact of climate stability on AM probability among angiosperms is insignificant, its direct impact is not. Thus, a 1% increase in the CSI resulted in a 0.8% reduction in AM probability. For freshwater fishes, a 1% increase in distance from the coastline results in a 1.2% decrease in AM probability. Conversely, latitude and ecological integrity increase resulted in a 1.5% and 0.3% increase in AM probability, respectively (Table 3). Finally, for terrestrial vertebrates, a 1% increase in latitude, distance from the coastline, and CSI resulted in 2.0%, 4.9%, and 1.6% reductions in AM probability, respectively. Conversely, a 1% increase in altitude and ecological integrity resulted in a 1.3% and 0.6% increase in AM probability, respectively (Table 4).

## Conservation status of areas with microendemic species

There is considerable variation in the conservation status of AMs across the region (Fig. 4). By using the 50% threshold, we found that 329 AMs have low ecological integrity and

**Table 2 Total, direct and indirect impacts of five local attributes on the probability of a hexagon harboring at least one microendemic angiosperm species in the Brazilian Atlantic Forest.**

| Attributes | Impact | Standard error | z | p |
|---|---|---|---|---|
| **Latitude** | | | | |
| Total | 0.336 | 0.763 | 0.441 | 0.659 |
| Direct | 0.232 | 0.482 | 0.480 | 0.631 |
| Indirect | 0.105 | 0.286 | 0.366 | 0.714 |
| **Altitude** | | | | |
| Total | 0.301 | 0.203 | 1.486 | 0.137 |
| Direct | 0.205 | 0.115 | 1.780 | 0.075 |
| Indirect | 0.096 | 0.109 | 0.882 | 0.378 |
| **Distance from the coastline** | | | | |
| Total | −1.738 | 0.481 | −3.610 | 0.000 |
| Direct | −1.194 | 0.242 | −4.934 | 0.000 |
| Indirect | −0.544 | 0.454 | −1.198 | 0.231 |
| **Climate stability index** | | | | |
| Total | 1.135 | 0.645 | 1.761 | 0.078 |
| Direct | 0.791 | 0.362 | 2.187 | 0.029 |
| Indirect | 0.345 | 0.366 | 0.942 | 0.346 |
| **Ecological integrity** | | | | |
| Total | 0.684 | 0.205 | 3.345 | 0.001 |
| Direct | 0.473 | 0.065 | 7.247 | 0.000 |
| Indirect | 0.211 | 0.187 | 1.126 | 0.260 |

conservation effort, 76 have high ecological integrity but low conservation effort, 37 have high ecological integrity and high conservation effort, and, finally, 32 have low ecological integrity and high conservation effort. Moreover, AMs representing all four categories are found across the region (Fig. 5).

## DISCUSSION

Mapping AMs shows that the Atlantic Forest is a large and complex biogeographic mosaic, where AMs represent islands or archipelagoes surrounded by transition areas having no microendemic species. In addition, microendemism is ubiquitous across the entire region. As such, our findings indicate a high regional biogeographic heterogeneity not reported previously (*Silva, Sousa & Castelletti, 2004*; *Nogueira et al., 2010*; *DaSilva, Pinto-da-Rocha & DeSouza, 2015*; *Silva et al., 2019*; *Peres et al., 2020*). Studies on microendemic species in other biodiversity hotspots have also documented highly complex mosaics of AMs and transition regions without microendemic species (*Kruckeberg & Rabinowitz, 1985*; *Wilmé, Goodman & Ganzhorn, 2006*; *Hobohm, 2013*; *Wulff et al., 2013*; *Caesar, Grandcolas & Pellens, 2017*), and such similarities may suggest that high internal biogeographical heterogeneity is a common attribute of all biodiversity hotspots.

We found that latitude, altitude, distance from the coastline, CSI, and ecological integrity can help distinguish AMs from non-AMs, but their impacts vary across

**Table 3 Total, direct and indirect impacts of five local attributes on the probability of a hexagon harboring at least one microendemic species of freshwater fishes in the Brazilian Atlantic Forest.**

| Attributes | Impact | Standard error | z | p |
|---|---|---|---|---|
| **Latitude** | | | | |
| Total | 1.51 | 0.73 | 2.09 | 0.04 |
| Direct | 1.68 | 0.60 | 2.80 | 0.01 |
| Indirect | −0.16 | 0.32 | −0.51 | 0.61 |
| **Altitude** | | | | |
| Total | −0.14 | 0.13 | −1.11 | 0.27 |
| Direct | −0.16 | 0.15 | −1.04 | 0.30 |
| Indirect | 0.02 | 0.04 | 0.39 | 0.70 |
| **Distance from the coastline** | | | | |
| Total | −1.18 | 0.31 | −3.76 | 0.00 |
| Direct | −1.30 | 0.32 | −4.06 | 0.00 |
| Indirect | 0.13 | 0.27 | 0.47 | 0.64 |
| **Climate stability index** | | | | |
| Total | 0.57 | 0.50 | 1.14 | 0.26 |
| Direct | 0.63 | 0.51 | 1.23 | 0.22 |
| Indirect | −0.06 | 0.12 | −0.49 | 0.63 |
| **Ecological integrity** | | | | |
| Total | 0.33 | 0.10 | 3.21 | 0.00 |
| Direct | 0.36 | 0.09 | 4.04 | 0.00 |
| Indirect | −0.04 | 0.07 | −0.48 | 0.63 |

taxonomic groups. For instance, distance from the coastline negatively correlates with AM probability in all taxonomic groups, and this variable has the highest total impact among angiosperms and terrestrial vertebrates. This is a relevant finding because distance from the coastline is not a geographic variable commonly measured and used in macroecological studies to predict both species richness and endemism, even though all global maps produced thus far demonstrate that, at least in some regions, coastal areas have a high density of species, and endemic species in particular (*Kier et al., 2009*; *Jenkins, Pimm & Joppa, 2013*). Thus, in the context of the Atlantic Forest, distance from the coastline possibly summarizes well the gradients of topographic complexity, rainfall, water shortages, temperature, and soils that distinguish the region's evergreen forests along the coastline from all the semideciduous and deciduous forests located in the region's inland (*Neves et al., 2017*; *Rezende et al., 2021*).

The hypothesis that there is a negative association between latitude and AM probability (*Saupe et al., 2019*) is accepted for terrestrial vertebrates but not freshwater fishes, as the association is positive in this group, requiring an additional explanation. Species richness and endemicity among neotropical freshwater fishes follow the core-periphery pattern, characterized by high species richness at the continental core and high species endemism at the continental periphery (*Albert, Petry & Reis, 2011*). Thus, while fish species diversity

**Table 4 Total, direct and indirect impacts of five local attributes on the probability of a hexagon harboring at least one microendemic species of terrestrial vertebrates in the Brazilian Atlantic Forest.**

| Attributes | Impact | Standard error | z | p |
|---|---|---|---|---|
| **Latitude** | | | | |
| Total | −2.01 | 0.76 | −2.66 | 0.01 |
| Direct | −1.90 | 0.88 | −2.17 | 0.03 |
| Indirect | −0.11 | 0.31 | −0.36 | 0.72 |
| **Altitude** | | | | |
| Total | 1.29 | 0.37 | 3.50 | 0.00 |
| Direct | 1.22 | 0.24 | 5.00 | 0.00 |
| Indirect | 0.07 | 0.22 | 0.32 | 0.75 |
| **Distance from the coastline** | | | | |
| Total | −4.89 | 1.30 | −3.76 | 0.00 |
| Direct | −4.62 | 1.08 | −4.28 | 0.00 |
| Indirect | −0.27 | 0.82 | −0.33 | 0.74 |
| **Climate stability index** | | | | |
| Total | −1.65 | 0.57 | −2.89 | 0.00 |
| Direct | −1.56 | 0.52 | −2.98 | 0.00 |
| Indirect | −0.09 | 0.26 | −0.33 | 0.74 |
| **Ecological integrity** | | | | |
| Total | 0.59 | 0.18 | 3.32 | 0.00 |
| Direct | 0.56 | 0.15 | 3.87 | 0.00 |
| Indirect | 0.03 | 0.10 | 0.33 | 0.74 |

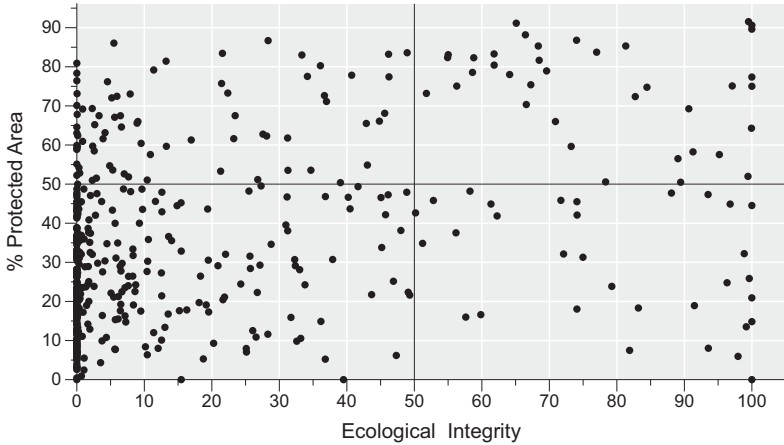

**Figure 4 The conservation status of areas with microendemic species in the Brazilian Atlantic Forest by assessing their percentage of protected areas and ecological integrity.**

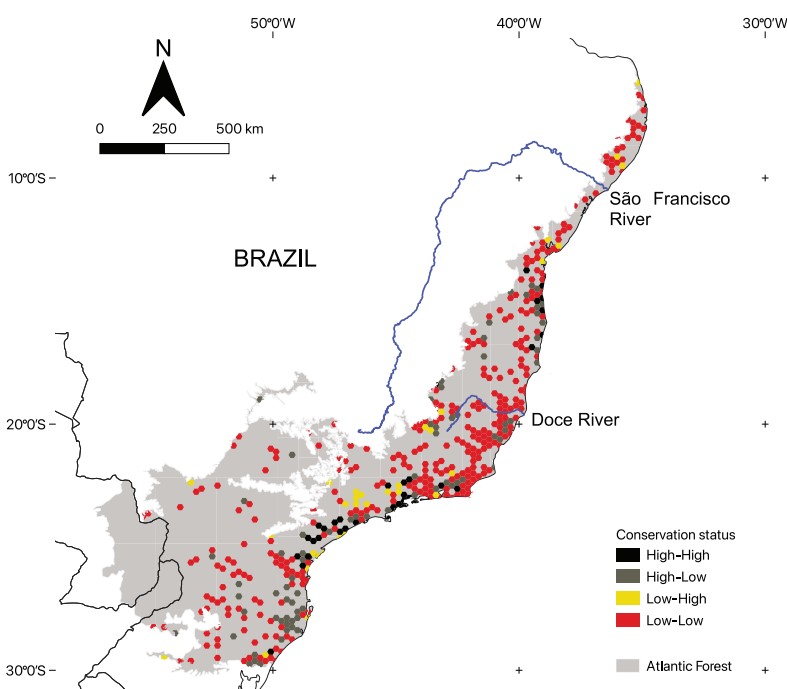

**Figure 5 The geographic distribution of the areas with microendemic species in the Brazilian Atlantic Forest according to their conservation status categories.**

decreases with latitude, endemicity does not. For instance, the area with the highest proportion of endemic fish species in the Atlantic Forest is in its south (*Albert, Tagliacollo & Dagosta, 2020*), showcasing a pattern not found among other vertebrates (*Peres et al., 2020*).

The hypothesis that altitude positively correlates with AMs is supported only for terrestrial vertebrates, so this relationship cannot be generalized to other taxonomic groups. This finding is unexpected because terrestrial vertebrates, given their dispersal capacity, are the least likely group among those we studied to be affected by topographic-driven isolation (*Steinbauer et al., 2016*). A potential explanation for this pattern is that, at least for angiosperms and freshwater fishes, altitude, instead of facilitating the maintenance of microendemic species, increases their extinction rates.

The hypothesis that long-term climate stability is positively associated with AMs is supported for angiosperms, matching the results of other studies (*Haffer, 1985*; *Fjeldså, Lambin & Mertens, 1999*; *Harrison & Noss, 2017*). However, against our predictions, this relationship between long-term climate stability and AM probability is negative for terrestrial vertebrates. This pattern, although unexpected, is not restricted to the Atlantic Forest. For instance, *Silva (1997)* reported that endemic species with small ranges in the Cerrado evolved in the region's most climatically unstable areas.

All three taxonomic groups support the hypothesis that ecological integrity is positively associated with AM probability, confirming that places with high levels of human disturbance are less likely to harbor microendemic species, with two hypotheses potentially

explaining this pattern: (a) such areas did house microendemic species, but they have since faced extinction due to human activities or (b) human activities are more intense in areas with attributes that do not facilitate the presence of microendemic species. Testing these two hypotheses formally can be a useful research endeavor for future biogeographic studies on the region. Our result reinforces the importance of considering human impact when examining endemicity patterns because local extinctions caused by human activities can distort general biogeographic patterns and lead to invalid conclusions. Considering anthropogenic impacts when studying species ranges is especially important in biodiversity hotspots, because these regions have lost 70% or more of their native primary vegetation (*Mittermeier et al., 2005*); consequently, the biogeographic patterns we observed herein may not represent precisely the biogeographic patterns that existed before the expansion of human activities across these regions.

The Brazilian Atlantic Forest has a large conservation gap necessitating closure to protect the AMs we identified in this article. Although they cover only 23% of the region, 69% have reduced ecological integrity and limited conservation efforts. As such, closing this gap requires concerted efforts toward implementing local and national strategies with policy instruments (*Silva et al., 2016*). Thus, we suggest three general actionable guidelines: (1) zero deforestation policies should be adopted for all AMs with native vegetation, (2) AMs with native vegetation and no conservation effort should be considered priorities for establishment as new protected areas, and (3) AMs with no native vegetation should be priorities for ecological restoration. Direct government action is limited and expensive, particularly because the private sector owns most of the Atlantic Forest (*Freitas et al., 2018*), so to ensure the region's long-term preservation, the private sector must establish a comprehensive network of private reserves that are carefully planned by using the most reliable scientific data available (*Silva, Pinto & Scarano, 2021*).

## CONCLUSIONS

Our results show that AMs in the Atlantic Forest are ubiquitous across the region, can be distinguished from non-AMs based on local attributes, and require urgent conservation actions. They also suggest that AMs are the product of complex interactions between the taxonomic groups' attributes and those of the locations in which they exist. As none of these attributes is stable, AMs are rendered dynamic—thus, once-widespread species could eventually become microendemic and microendemic species can eventually become widespread. Conclusively, the diverse relationships between AMs and their ecological, historical, and anthropogenic attributes across different taxonomic groups should be considered the norm rather than the exception in biogeographic studies of the Atlantic Forest and other large biogeographical regions.

We recognize that some of the AMs we identified could be sampling artifacts, because knowledge of the Atlantic Forest's biota is still lacking. Regardless, the Atlantic Forest represents Brazil's most well-sampled region, harboring the country's highest density of scientists and organizations focused on biodiversity (*Silva et al., 2016*). To mitigate the influence of incomplete knowledge, we used a conservative approach by considering only species records with documented and peer-reviewed evidence in our analyses. While

additional data may reveal more extensive ranges for some of the species included in our analysis, we believe the general patterns described here are robust enough to both offer insights into the biogeography of one of the most important biodiversity hotspots globally and create a foundation for comparative studies using other tropical forest regions.

## ACKNOWLEDGEMENTS

We thank Luis Barbosa for helping us with the maps and Julie Topf for reviewing the first draft of the manuscript. We are grateful to Maria Alice S. Alves, Caio C. Missagia, and an anonymous reviewer for their valuable suggestions to enhance the manuscript.

### Funding

José Maria Cardoso da Silva is supported by the University of Miami and the Mycorrhiza Fund. Helder F.P. de Araujo and Célia C.C. Machado are supported by the Conselho Nacional de Desenvolvimento Científico e Tecnológico. The funders had no role in study design, data collection and analysis, decision to publish, or preparation of the manuscript.

### Grant Disclosures

The following grant information was disclosed by the authors:
University of Miami and the Mycorrhiza Fund.
Conselho Nacional de Desenvolvimento Científico e Tecnológico.

### Competing Interests

José Maria Cardoso da Silva is an Academic Editor for PeerJ.

### Author Contributions

- Helder F.P. de Araujo conceived and designed the experiments, performed the experiments, analyzed the data, prepared figures and/or tables, authored or reviewed drafts of the article, and approved the final draft.
- Célia C.C. Machado performed the experiments, analyzed the data, prepared figures and/or tables, authored or reviewed drafts of the article, and approved the final draft.
- José Maria Cardoso da Silva conceived and designed the experiments, performed the experiments, analyzed the data, prepared figures and/or tables, authored or reviewed drafts of the article, and approved the final draft.

### Data Availability

The list of microendemic species and their records used in the analyses are available in the Supplemental File.

### Supplemental Information

Supplemental information for this article can be found online at http://dx.doi.org/10.7717/peerj.16779#supplemental-information.

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
