# Peer review of "The distribution and conservation of areas with microendemic species in a biodiversity hotspot: a multi-taxa approach"

_PeerJ, doi:10.7717/peerj.16779_

## Round 0.1 · original submission · Major Revisions

Dear Dr. Silva,

Based on the decision letters provided by three reviewers and their valuable insights for improvements, I consider that your manuscript may be accepted for publication the next time it is submitted for evaluation.

Please consider all issues raised by all three reviewers and prepare a rebuttal letter informing the point-by-point improvements you add to the text in order to correct the issues raised. Otherwise, please respond to the reviewers the reason why this or that change was not performed.

Sincerely,
Daniel Silva

Reviewer 1 ·

Basic reporting

The manuscript complies with all the criteria, it is clear and well-organized thorough.

Experimental design

1) The manuscript is a good fit to PeerJ, presents an interesting question and provides a new look into microendemics in the Brazilian Atlantic Forest, a biodiversity hotspot.
2) There is an issue with one of the presented research questions that I suggest should be carefully considered. I provide more details on it in the additional comments section.
3) The methods are clear and described with enough information to be replicated together with the raw data that accompanies the manuscript.

Validity of the findings

The microendemics species occurence data has been provided and additional data and manipulations are well described in the methods. It is statistically sound, however I provide some comments in the "additional comments" section that are related to some caveats of the data analyses.

Additional comments

Abstract. Only three of the four variables are mentioned, it is missing the climate stability variable.
Author affiliation: affiliation coding does not match coding institutional and correspondence addresses.

Introduction
Lines 62-64: I suggest that the "smallest spatial unit" is best explained? Does it correspond to species range? Or, is it a variable defined depending on the study? If the latter, it sounds like it is too variable or flexible to be manageable or applicable to biogeography, conservation, and management.
Line 103: remove "to protect them"
Line 106: Why is it that only reactive strategies are possible within hotspots?
Lines 125-129: I do not agree with the last statement. There certainly is a higher probability of being found within AEs (as shown in the study) based on how these are drawn, but it does not precludes the occurrence of endemic species outside of AEs.

Methods:
Line 152: Please, could you be specific about what sort of data was unavailable?
Lines 154-155: Add 'et al.' to the citation. Peres et al. (2020) recognize five AEs for the Atlantic Forest which are then considered as three areas in the manuscript. It needs to be explained further as to why it was reduced to three. Also, as expected AEs do not overlap perfectly among the studied taxonomic groupings. Thus, why not using taxonomically specific AEs for the analyses? I suggest that all those decisions about how the data was treated be clear to the reader.
Lines 177-178: This is the first time that four subregions are mentioned. Please, explain what they are and how they were used.
Lines 180-183: This is one of main concerns regarding the manuscript and that I think require some more thought and justification.There seems to be a circular reasoning regarding this analysis, as AEs are draw based on the distribution of endemics, likely including the MAs evaluated in the manuscript. Therefore, I am not sure if this analyses is relevant to this study or scientifically sound as it is expected that MAs will be within AEs.
Lines 214-217: However, as every hexagon is only coded by presence/absence of endemics, it does not account for species richness which would be an indicative of clustering. Is there an alternative way of exploring clustering using a richness proxy, instead of presence/absence?

Results:
Lines 256-258: I am curious about the correlation among explanatory variables, specially between latitude and climatic stability. I suggest testing if explanatory variables are not correlated, maybe using the variance inflation factor (VIF)?
Lines 271-272: Here, I am also thinking about the correlation between explanatory variables (vegetation cover ~ protected area cover), which I also suggest that should be tested.

Discussion:
Lines 281-286:
It goes back to what I mentioned above, there is a some circularity in these thoughts and the definitions of AEs and MAs. Since AEs were draw based on the overlap among species distributions, which can considers the range and occurrence of (micro)endemics and considering many hierarchical levels (as mentioned in the introduction), I am not sure if this statistical analysis is relevant and/or sound.
Line 292: I think that accounting only for the presence/absence of species in a given grid can lead to incomplete assessments of current MAs patterns. I suggest that considering the total number of microendemic species would be more relevant than the presence or absence of MAs within map grids. Yet, I am not saying that the presence of these microendemics outside of AEs isn't relevant from a biogeographic and conservation perspective.
Lines 297-299: I believe this might be a good justification for this analysis (despite the circularity and relationship between AEs and MAs). In addition, the study highlights the occurrence of MAs outside of protected areas, which, from a conservation perspective, is more important than their occurrence outside of AEs. Although AEs are by definition the overlap of many endemic species, it does not preclude the occurrence of endemics outside of AEs.
Lines 305-307: This result is not shown in Table 2 or described in the results.

Supplementary material:
I suggest adding a table description within the raw data file.

·

Basic reporting

This work is a very good contribution to better understand microendemism in the Atlantic forest, one of the most important global biodiversity hotspots. It provides insights into the biogeography of hotspots and it will be important for comparative studies with other tropical hotspots. In general, it is well written, with appropriate background context and methods. The results are well presented, with relevant figures, with very good quality. The discussion could be deeper; sometimes the results are repeated and somepoints could be better explored, such as the variable distance to the coast. The recommendations are interesting, but sometimes not directly supported by the results.

Experimental design

In general the methods are described with sufficient detail. However, about the used species datasets, it is important to inform whether the database was validated and, if so, how it was done, as sometimes there are errors or outliers.

Validity of the findings

Conclusions/recommendations sometimes are not limited to supporting results. About the species datasets is important to inform whether the database was validated and, if so, how it was done.

Additional comments

Specific comments:
Page 10, Line 274: replace “3” with “three”
Page 16, Lines 309-314: The distance from the coastline could be better discussed. This variable could be a proxy for ther variables such as humidity and precipitation, for example.

Page 15, Paragraph starting in Line 316: The result is being repited here as in other points of the discussion. It would be better to change for something to be discussed more directly with each point, such as:
"the negative association between latitude with AMs for terrestrial vertebrates matches that expected from the model proposed by Saupe et al. (2019)...”

Page16, Paragraph starting in Line 346: The recommendations are interesting, but some of them are not supported directly by the results of the present work.

·

Basic reporting

The manuscript presents a relevant study that deals with microendemism of groups of plants, freshwater fish and some terrestrial vertebrates in the Brazilian Atlantic Forest. Using geospatial analyses, the authors assessed whether the number of microendemisms in the recognized endemic regions of the Atlantic Forest is greater than "expected", in order to highlight more important locations for conservation. Also describe the occurrence of microendemisms as a function of geographic variables, specifically latitude, altitude, distance from the coastline, and climatic stability index, in order to generate predictive models for the occurrence of microendemisms. Finally, they assessed the presence of vegetated and protected areas in areas where microendemisms occur.
The study addresses a current theme and brings some novel results, which in a way complement previous studies with regions of endemism and microendemic species in the Atlantic Forest in Brazil.
Some general comments are presented here, and specific ones are in the attached pdf. I hope this review can help the authors to further improve the work, which is very interesting.

Introduction
The introduction is written in accordance with scientific language and the references are appropriate for the topic covered in the aforementioned paragraphs. However, I missed addressing the relevance of the study in relation to the ecological theory to which I understand the topic "microendemism" fits, the Niche Theory. I understand that the authors are not dealing with a niche, but it is necessary to contextualize the study theoretically. In fact, there is experimental evidence that the range limits are in fact the limits of the niche (e.g. https://doi.org/10.1111/ele.12604). However, in none of their conceptions, whether by Grinnell or Elton, or Hutchinson's multidimensional hypervolume, the niche theory was presented by the authors. I believe that at least the possibility of overlapping range with niche can be addressed, if the authors agree. The variables measured by the authors are related to the Grinnellian niche. I made comments in the attached pdf.

Experimental design

The Experimental design is suitable for the proposed objectives. Statistical hypotheses are appropriate for testing the scientific hypotheses formulated by authors (Lines 180, 218). However, I recommend that a brief description of the interpretation of the results of hypothesis tests (180-188) and (208-219) be included so that readers unfamiliar with spatial analyzes can interpret the results. It is important to consider that this study could encourage decision makers to take conservation actions. Therefore, a wide readership is expected.

The study uses the known range of microendemic species (line 158) and updates them based on data from citizen science platforms (lines 167-169). Regarding updating, the authors did not consider including databases of recent works relevant to the study's target groups. I strongly recommend that it be included. It is a database made up of data from researchers who have been working for decades throughout the Brazilian Atlantic Forest (see references in the attached pdf). Although the authors cite the references of the bases used (lines 163-166),

it is not clear whether the polygons used deal with range or niche. In fact, there is experimental evidence that the range limits are in fact the limits of the niche (e.g. https://doi.org/10.1111/ele.12604). It's important to put it in context.

The group "seed plants" is too broad to be treated as unique (line 161). Is this common in literature? If so, cite different references. Even so, I understand that it is more explanatory to create subgroups. Here are organisms with a life cycle of a few months to thousands of years, as small as microorchids to the large trees of the Atlantic Forest. I recommend that authors consider adopting subdivisions. This is necessary to direct conservation actions. I made suggestions in the attached pdf.

Validity of the findings

Results

In general, the results are consistent with the objectives and methodology. I believe that the presentation of results can be improved by clearly informing when the hypotheses were or were not corroborated. See comments in the attached pdf.

Forgive me if I didn't understand, but I couldn't locate the results of microendemisms overlapping by hexagon. In other words, I was unable to identify the hottest places in terms of occurrence of microendemisms. The authors indicate that they counted the number of microendemic species in each exagon used in the spatial analyzes (lines 175-176). However, Figure 2 only shows the presence and absence of microendemic species per exagon. I suggest that the figure be redone considering the quantitative values per exagon. This way, it will be possible to identify the most relevant areas (with overlapping richness) and discuss these results in the conservationist context adopted by the authors.

If possible, the results for "plants with seeds" should be re-presented considering subgroups. For example, herbs-shrubs-trees or terrestrial-rupicola-epiphyte, or some other subgroup that has some biological/ecological significance. I made suggestions in the attached pdf.

Conclusions

The conclusions are in accordance with the general structure presented in the manuscript. As a suggestion, I recommend removing some sentences from the beginning of the item, which have an introduction or results tone (see pdf comments). If adjustments are made that are recommended by me or another reviewer, it is important to remember to make the respective adjustments here.

---

## Round 0.2 · Minor Revisions

Dear Dr. Silva,

In this new review round, two reviewers indicated very minor changes are needed before accepting your manuscript. Once you proceed with such changes, I believe your manuscript will be formally accepted for publication in PeerJ.

Sincerely,
Daniel Silva.

Reviewer 1 ·

Basic reporting

No additional comment from previous round of review.

Experimental design

It is stated several times throughout the manuscript that the authors identify attributes that set apart AMs from non-AMs. I don't think it is what the authors have achieved or analyzed, but I emphasize that I am not an expert on the methods used. The authors have found attributes that are correlated with the occurrence of AMs, but it does not mean that those attributes set AMs and non-AMs apart. Especially because AMs are surrounded by non-AMs that are likely to have very similar attributes, but also have more variability in these attributes than AMs by being more widespread and common. Interestingly, the idea of setting AMs and non-AMs apart is almost absent from the results and discussion, where the focus is on the correlations.

Validity of the findings

No additional comment from previous round of review.

Additional comments

Line 120: remove "Field"
Lines 238-239: remove "(a) AMs with high ecological integrity and high conservation effort;"
Line 260: remove "In contrast". I do not think the results are contrasting, as there is only a single attribute between the models for angiosperms and fishes.
Lines 310-313: Would these features be the result of geographical complexity and features, e.g., Serra do Mar, Mantiqueira, etc?
Line 333: "Result" is repeated twice, please rephrase.
Lines 341-342: Was the pattern driven by extinction of microendemics, or are human settlements/activities located in regions that are less likely to harbor microendemics?
Line 351: Add "Forest"
Line 389: Change "referee" for "reviewer"

Also, at tables and most of the text, I suggest changing the usage of "place attributes" for "local/hexagon/cell attributes" or other term of choice.

Supplemental File: it is incomplete, the authors might have submitted the wrong file.

·

Basic reporting

The authors made substantial changes in the manuscript, which improved it very much. I consider that it can be accepted for publication after few changes mentioned below:

Page 13, L367-368: remove one one the item (a), which is duplicated.
Page 13, L384: replace "8" with "eight"
Page 20, L600: replace "Maria Alice Alves" with "Maria Alice S. Alves"
remove one of the items (a), which is duplicated.

Experimental design

Good. No comment.

Validity of the findings

Good. No comment.

Additional comments

The authors made substantial changes in the manuscript, which improved it very much. I consider that it can be accepted for publication after few changes mentioned below:

Page 13, L367-368: remove one one the item (a), which is duplicated.
Page 13, L384: replace "8" with "eight"
Page 20, L600: replace "Maria Alice Alves" with "Maria Alice S. Alves"
remove one of the items (a), which is duplicated.

·

Basic reporting

no comment

Experimental design

no comment

Validity of the findings

no comment

Additional comments

no comment

---

## Round 0.3 · accepted · Accept

Dear Dr. Silva,

I am pleased to inform you that your manuscript has been formally accepted for publication in PeerJ.

Sincerely,
Daniel Silva

Reviewer 1 ·

Basic reporting

Nothing to add. Congratulations on your work and all the best with future research.

Experimental design

Nothing to add.

Validity of the findings

Nothing to add.

Additional comments

Nothing to add.